# Metabolic Indicators and Emotional Distress Symptoms Related to Overweight in Youth: A Brief Network Analysis

**DOI:** 10.3390/healthcare13233096

**Published:** 2025-11-27

**Authors:** Lesly Bucio-Palma, Lina García-Mier, Martha Cruz-Soto, Angela Vargas-Rodríguez, Jorge Palacios-Delgado

**Affiliations:** 1Unidad de Investigación en Neurociencias, Facultad de Medicina, Universidad del Valle de México, Campus Querétaro, Santiago de Querétaro 76230, Mexico; dra.leslybuciop@gmail.com; 2Escuela de Ciencias de la Salud, Campus Querétaro, Universidad del Valle de México, Santiago de Querétaro 76230, Mexico; lina.garcia@uvmnet.edu (L.G.-M.); a110118863@my.uvm.edu.mx (A.V.-R.); 3Coordinación Nacional de Investigación en Ciencias de la Salud, Universidad del Valle de México, Campus Querétaro, Santiago de Querétaro 76230, Mexico; martha.cruzso@uvmnet.edu; 4Unidad de Investigación en Neurociencias, Facultad de Psicología, Universidad del Valle de México, Campus Querétaro, Santiago de Querétaro 76230, Mexico

**Keywords:** overweight, metabolic indicators, emotional distress, waist-to-hip ratio, happiness and network analysis

## Abstract

Overweight and obesity in young people represent a major public health challenge, not only due to their association with metabolic alterations but also because of their impact on emotional well-being. **Background/Objectives**: The objective of this study was to determine the structure of the network between metabolic indicators and emotional symptoms **Methods**: A sample of 78 university students was assessed through biochemical, anthropometric, and psychometric measures. **Results**: A total of 37.9% showed risk levels associated with excess weight and moderate emotional distress. There exist significant links between central adiposity and lipid alterations, as well as positive correlations between anxiety, depression an emotional exhaustion and inverse with happiness. **Conclusions**: The results highlight an interaction between metabolic and emotional factors even among individuals with normal weight, underscoring the value of network models for early risk detection. The findings are useful for implementing preventive strategies in university settings to promote improved health and emotional well-being.

## 1. Introduction

An increase in body weight is associated with an imbalance in the body [1,2]. Globally, in 2022, more than 390 million children and adolescents aged 5 to 19 were overweight [3]. Currently, Mexico ranks second in global obesity rates, with nearly 40% of its population affected [4]. Overweight and obesity are closely linked to the development of non-communicable diseases such as type 2 diabetes, hypertension, and cardiovascular diseases.

Previous research [3,5,6]; highlights that certain biochemical markers—such as elevated fasting glucose levels combined with obesity—can lead to insulin resistance (IR) or type 2 diabetes [7,8] as well as an increase in lipid profile components, including triglycerides and cholesterol, which may be altered by the accumulation of body fat [9]. These biochemical alterations are linked to insulin and contribute to hypertriglyceridemia and chronic inflammation. The above-mentioned metabolic alterations —are not often noticeable in early stages of metabolic and cardiovascular diseases. Therefore, it is essential to evaluate anthropometric and biochemical parameters in individuals with excess body weight, using indicators such as body mass index (BMI) [10] and waist-to-hip ratio (WHR), which assesses the distribution of body fat [11].

Obesity presents a series of complications that affect an individual’s psychological state, such as anxiety and depression [12]. Exist studies have found that metabolic imbalance, associated with being overweight and obesity, significantly increases cardiovascular risk. This imbalance not only affects the cardiovascular system but can also influence mood regulation, as chronic inflammation and endothelial dysfunction are linked to the development of depressive symptoms. This suggests a bidirectional mechanism between obesity and mental well-being, where both factors reinforce each other [13]. Obesity has been observed to disrupt emotional regulation, increasing vulnerability to disorders such as depression and anxiety [14]. This evidence suggests that psychological effects of obesity are not only a consequence of excess weight but also act as a contributing factor to the disease. In this regard, Frank et al. [15] found that obesity is associated with a specific cluster of depressive symptoms, and that these associations are explained by systemic inflammation, further supporting the bidirectional interaction between metabolic and psychological factors in obesity.

### 1.1. Depression Associated with Overweight and Obesity

Several studies have demonstrated a relationship between depression and obesity. Data showed that individuals with depression are 37% more likely to develop obesity, while those with obesity have an 18% greater chance of experiencing depression. Blaine et al. [16] reported that individuals with depressive symptoms are at greater risk of developing obesity in the future, with the risk being particularly elevated among adolescent females. Research by Luppino et al. [17] indicated that the relationship between depression and obesity is bidirectional; in other words, depression may lead to obesity, and excess weight may increase the likelihood of developing depressive symptoms. Palomino-Pérez et al. [18] described that many individuals with obesity use food as a coping mechanism for negative emotions, which in turn contributes to further weight gain. Jantaratnotai et al. [19] emphasized that the co-occurrence of depression and obesity increases the risk of developing either condition and highlighted that treating one may positively influence the course of the other.

### 1.2. Stress and Anxiety Associated with Overweight and Obesity

Stress has been identified as a factor that disrupts self-regulation, leading to the consumption of high-calorie foods and making weight control more challenging [20]. Similarly, Metz et al. [21] reported an association between stress and reduced quality of life among overweight and obese individuals. More recent findings indicate that overweight individuals report higher levels of stress [22].

Numerous studies [23,24] have demonstrated a close relationship between overweight and obesity with stress and anxiety. Research conducted among Mexican adolescents [25] revealed that those individuals with overweight or obesity experience significantly higher levels of anxiety and stress compared to peers with a healthy weight. Similarly, in a sample of Mexican adults, a significant correlation was observed between obesity and the presence of anxiety [23]. Furthermore, chronic stress has been identified as a risk factor in both the development and persistence of non-communicable diseases such as obesity, as it interferes with emotional regulation and encourages unhealthy eating behaviors [24].

### 1.3. Positive Emotions Associated with Overweight and Obesity

Research on the emotional state associated with individuals with excess weight has traditionally focused on negative emotional states, while positive emotions such as happiness have been less studied. In this regard, Andrei et al. [26] found that individuals with obesity (BMI ≥ 40) tend to experience lower levels of happiness compared to individuals with normal weight. Similarly, Saxena et al. [27] identified a negative relationship between obesity and happiness, indicating that obesity can negatively affect an individual’s level of happiness. Along the same lines, Katsaiti et al. [28] pointed out that happiness and stress are interrelated in patients who have excess weight and obesity, influencing both their psychological and physical well-being.

In addition, Diener et al. [29] emphasized that happiness plays a protective role against psychological distress, reporting that individuals with higher levels of happiness exhibit lower levels of depression, emotional exhaustion, and stress. These findings suggest that the emotional impact of obesity extends beyond happiness and involves a broader spectrum of psychological symptoms, including depression, stress, and anxiety, all of which affect individuals’ overall well-being. Kok et al. [30] found that emotional well-being is associated with lower blood glucose levels. SimilarlyTheoretical mechanisms that underlie the association between loneliness and health are also described [31]. Regarding lipid profiles, Toker et al. [32] identified that positive affect is associated with lower triglyceride levels, while Steptoe et al. [33] observed that individuals with greater emotional well-being exhibit lower total and LDL cholesterol levels. This evidence suggests that happiness not only protects against psychological distress but may also play a regulatory role in metabolic indicators.

## 2. Current Study

This study addresses a topic of high relevance in the field of public health [34], as inadequate dietary preferences [35] and excess weight are highly prevalent. Understanding the interaction between metabolic and emotional issues affecting individuals with excess weight—including those of normal weight who may still be at risk—is essential, as these conditions not only affect quality of life but also place a significant burden on healthcare systems.

This research contributes new knowledge regarding the interaction between metabolic and emotional issues in young individuals considered healthy [36]. Existing studies [12,37] have predominantly focused on patients who have excess weight, often overlooking normal-weight youth who appear healthy but may already present underlying metabolic or emotional disorders without obvious symptoms [38].

Previous research [6,9,24,26] has independently addressed the relationship between metabolic markers and certain emotional symptoms, both negatively and positively. Although these studies have explored the impact of emotional symptoms in individuals with excess weight, they primarily employ linear statistical methods, which limit a deeper understanding of their interaction—particularly when examining samples of normal-weight youth who present elevated metabolic indicators.

Currently, there is limited evidence examining the bidirectional interaction [15,19,37] between metabolic markers and emotional manifestations. In this context, non-linear approaches, such as artificial neural networks [39], offer an innovative alternative to model and understand the complex dynamics among multiple factors that may be associated with excess weight. Therefore, the objective of this research was to determine the connectedness network of different symptoms, and potential bridge symptoms to classify patterns and predict adverse emotional states from metabolic data in young university students.

Based on the background information identified in previous studies [40,41], we propose that the architecture of an artificial neural network will reveal a connectivity structure between specific metabolic indicators and the manifestation of positive and negative emotional symptoms (anxiety, depression, and emotional exhaustion). Specifically, we hypothesize that depression and anxiety symptoms will exhibit high centrality, and that emotional exhaustion symptoms will be the most central among anxiety symptoms. We predict that emotional exhaustion will serve as a bridge symptom between anxiety and depression. We expect happiness to be inversely linked with emotional exhaustion, depression, and anxiety symptoms. Furthermore, we hypothesize the existence of bridge symptoms through connections with different metabolic patterns, influencing the co-occurrence of multiple positive and negative emotional symptoms.

## 3. Method

### 3.1. Design Research

An associative cross-sectional study was conducted during the year 2024, including first-year students entering their second semester at a private university in the city of Querétaro, Mexico, who agreed to participate in the research.

For the sample size calculation, a total population of 497 first-year students enrolled in health sciences in February 2024 was considered and a non-probabilistic sampling method was used. A 95% confidence interval with a 5% margin of error was applied, resulting in a sample of 217 participants. Students who did not meet the requirement of fasting for at least eight hours prior to blood sample collection, which prevented the acquisition of biochemical indicators—were excluded. A second exclusion criterion was the incomplete completion of the psychological measurement instruments.

Finally, 78 undergraduate students participated in the study, 27 men and 51 women, between 17 and 32 years old (M = 19.73; SD = 3.1). Of these, 75.67% were veterinary students, 12% were medical students, 12.16% were biomedical students, and 2.70% were biotechnology students. The study was conducted from January to December 2024.

### 3.2. Measurements

#### 3.2.1. Biochemical Evaluations

Samples were taken using the portable diagnostic device Accu-Chek Accutrend Plus (Roche Diagnostics, Basel, Switzerland) to measure cholesterol and triglyceride levels, and the Accu-Chek Instant device (Roche Diagnostics, Basel, Switzerland) was used to measure glucose levels.

Glucose: Analyses were stratified according to risk levels based on reference values established by the American Diabetes Association et al. [42], as follows: (a) Normal or Risk Level 1 (less than 100 mg/dL), (b) Prediabetes or Risk Level 2 (between 100 and 125 mg/dL), and (c) Diabetes or Risk Level 3 (greater than 126 mg/dL).Cholesterol: Analyses were categorized into different risk levels based on reference values established by the Mexican Social Security Institute et al. [43], as follows: (a) Desirable or Risk Level 1 (less than 200 mg/dL), (b) Borderline High or Risk Level 2 (between 200 and 239 mg/dL), and (c) High or Risk Level 3 (greater than 240 mg/dL).Triglycerides: Data were organized according to risk level, using the reference criteria from the Mexican Social Security Institute et al. [43], as follows: (a) Normal or Risk Level 1 (less than 150 mg/dL), (b) Borderline High or Risk Level 2 (between 150 and 199 mg/dL), (c) High or Risk Level 3 (between 200 and 499 mg/dL), and (d) Very High or Risk Level 4 (greater than 500 mg/dL).

#### 3.2.2. Anthropometric Measurements

Body weight and height were measured using the Omron Full Body Sensor BF300 scale (Omron Corporation, Osaka, Japan), and waist and hip circumferences were measured with a non-stretchable flexible measuring tape. Based on these measurements, the following indicators were obtained:Body Mass Index (BMI): Classified according to the Centers for Disease Control and Prevention et al. [44], as follows: (a) Underweight or Risk Level 1 (below 18.5), (b) Normal weight or Risk Level 2 (between 18.5 and 24.9), (c) Overweight or Risk Level 3 (25.0–29.0), and (d) Obesity or Risk Level 4 (30.0 or higher).Waist-to-hip Ratio (WHR): Categorized according to risk level based on the criteria established by Rosas et al. [45], as follows: (a) Low health risk or Risk Level 1 (<0.95 in men and <0.80 in women), (b) Moderate risk or Risk Level 2 (81–85 in women and 96–100 in men), and (c) High risk or Risk Level 3 (>86 in women and >100 in men).

#### 3.2.3. Patient Health Questionnaire-9 for Depression (PHQ-9)

The PHQ-9 is a screening scale that measures the presence and severity of depressive symptoms Kroenke et al. [46] employs a 4-point Likert scale using a range from 0 (not at all) to 3 (nearly every day), with total scores ranging from 0 to 27 points (9 items). A cut-off score of 10 points is recommended for evaluation [47]. In the current study, internal consistency reliability (α = 0.87 [CI95% = 0.86–0.89]) was adequate. Additionally, the CFA indicates adequate construct validity (CFI = 0.97, TLI = 0.96, NFI = 0.95, IFI = 0.97, RMSEA = 0.06). A Spanish-language version has been used [48].

#### 3.2.4. Self-Assessment of Happiness

A modified version of the self-assessment happiness scale proposed by Abdel-Khalek et al. [49] was used. For the purposes of this study, a single-item measure was applied, asking participants to evaluate their level of happiness over the past 30 days using a numerical rating scale from 1 to 10, displayed horizontally with equal intervals. Participants were instructed to consider 1 as the minimum score and 10 as the maximum, and to write the number that best represented their feelings. The scale demonstrates a one-week test–retest reliability of 0.86, indicating high temporal stability. The item has content validity, concurrent validity with positive emotions and divergent validity with negative emotions. It has been used in previous studies conducted in Mexico [50], yielding excellent results. In the present study, the following classifications were applied: scores ≤ 7 indicated low happiness, score 8 indicated moderate happiness, and scores ≥ 9 indicated high happiness. The cut-off point was established at 8 (M + 0.5 SD). 

#### 3.2.5. GAD-7 (Generalized Anxiety Disorder 7-Item)

The GAD-7 scale is a brief instrument with four Likert-type response options: “not at all,” “several days,” “more than half the days,” and “nearly every day,” scored as 0, 1, 2, and 3, respectively [51]. The scale is reliable (α = 0.93) for assessing symptoms of generalized anxiety and shows concurrent validity with other similar scales [52]. The total score is subdivided into three categories: mild anxiety, moderate anxiety (10), and severe anxiety. A score of 10 or higher is used to identify the presence of anxiety symptoms. For the purposes of this study, we used this cut-off point. In the current study, internal consistency reliability (α = 0.88 [CI95% = 0.85–0.90]) was adequate. Additionally, the CFA indicates adequate construct validity (CFI = 0.99, TLI = 0.98, NFI = 0.98, IFI = 0.99, RMSEA = 0.04).

#### 3.2.6. Emotional Exhaustion Scale (EES)

This scale is used to measure emotional exhaustion or burnout as an initial response to stress [53]. It contains 10 statements about emotional exhaustion. The answers are based on a Likert scale with evaluation from never = 1 to always = 5, considering the last 12 months of student activity. The scale has concurrent validity with anxiety and an internal structure with values of CFI = 0.90; GFI = 0.89; NFI = 0.88; RMSEA = 0.11. The authors report that Cronbach’s Alpha coefficient value = 0.90. The scale offers values ranging from 10 to 50 points. A score of 26 indicates moderate emotional exhaustion, and a score of 42 indicates high emotional exhaustion.

## 4. Procedure

Participants were approached to inform them about the research, the objectives of the study, and its benefits. After receiving this information, they were provided with an informed consent form, ensuring that they understood all the details of the study and gave their explicit consent. Once the consent form was signed, data collection began through a digital questionnaire on Google Forms, which included emotional status. The data were recorded with an emphasis on the confidentiality of the information provided. Subsequently, participants were scheduled for blood pressure measurement, which was conducted in a calm environment, with each participant seated and resting for at least five minutes prior to the measurement, with the arm supported at heart level. Systolic and diastolic pressure values were recorded following standardized protocols for cardiovascular assessment. For the collection of biochemical indicators, a capillary blood sample was obtained in the morning after a minimum of eight hours of fasting. Glucose, cholesterol, and triglyceride concentrations were determined. Subsequently, anthropometric measurements were carried out according to the standardized procedures of the National Center for Preventive Programs and Disease Control.

## 5. Statistics Analysis

Descriptive statistics were used, with measures of central tendency and dispersion, for metabolic indicators and emotional symptomatology. The Shapiro–Wilk Test was used to test the normal distribution. Spearman Rho’s correlation was used to assess the relationship between emotional symptoms, anthropometrics measurements and metabolic indicators. Finally, we used network analysis to evaluate the interactions of anthropometric measurements, metabolic indicators and negative and positive emotional symptoms.

For the network analyses of metabolic indicators and emotional symptoms, we used JASP Program version 0.19.1 (JASP Team, Amsterdam, The Netherlands, 2024). Each variable represents a node and an association between a node is referred to as an edge (line). To visualize the network structures blue edges, represent positive associations between two nodes, whereas red edges represent negative associations. We used correlations estimator to determine the associations between metabolic target variables and emotional constructs and bootstrapping routines implemented in the package to gain information on the precision of parameter estimates. Furthermore, we calculated four indexes of centrality (strength, closeness, betweenness and influence) [54]; to qualify the importance of each node, we calculated node centrality to identify which symptoms are most central to the network and which indicators and symptoms are most central to the network [55].

## 6. Results

### 6.1. Descriptive Statistics of the Study Variables

Body weight and height were converted into BMI, revealing that most participants were classified as having normal weight (48.6%), followed by those who were overweight (23.0%), obese (14.9%), and underweight (13.5%). The sample exhibited a healthy BMI range, while 37.9% showed risk levels associated with excess weight. According to the waist-to-hip ratio, participants were classified as low risk (77.8%), followed by moderate risk (12.5%) and high risk (9.7%). Table 1 shows glucose values within normal ranges among participants. Cholesterol levels suggest some variability but remain within desirable parameters. In contrast, triglyceride concentrations show a wide dispersion, with a tendency toward elevated values.

Regarding psychological factors, depressive symptoms indicated that most participants presented either mild intensity symptoms (36.5%) and minimal depressive symptoms (35.1%). Cases of high (8.1%) or severe depression (1.4%) were infrequent within the sample. Anxiety levels showed that most participants reported symptoms with low frequency (39.2%) and moderate frequency (37.8%), followed by high (16.2%) and, to a lesser extent, severe levels (6.8%). In terms of emotional exhaustion, 35% of participants showed high symptom levels, followed by moderate (27%) and low (22%) levels. Finally, an analysis of happiness levels revealed that average scores were below the established cutoff point of 8.

### 6.2. Correlation Analysis

Statistically significant correlations were found between metabolic and anthropometric variables. BMI showed only a moderate positive correlation with glucose levels (*r* = 0.35, *p* = 0.00), suggesting that higher blood glucose levels are associated with higher BMI. This relationship could indicate a possible trend toward insulin resistance in individuals with elevated BMI. WHR risk presented positive correlations with total cholesterol (*r* = 0.27, *p* = 0.03) and triglycerides (*r* = 0.31, *p* = 0.01), but not with glucose levels.

Statistically significant associations were found among psychological variables. Anxiety showed a strong positive correlation with depression (*r* = 0.807, *p* < 0.001) and with emotional exhaustion (*r* = 0.63, *p* < 0.001). Depression was also positively correlated with emotional exhaustion (*r* = 0.66, *p* < 0.001). In contrast, happiness levels showed negative correlations with anxiety (*r* = −0.39, *p* < 0.001), depression (*r* = −0.46, *p* < 0.001), and emotional exhaustion (*r* = −0.32, *p* = 0.00), suggesting that increases in emotional distress indicators are associated with lower levels of happiness.

### 6.3. Neural Network Structure

Network analysis used to estimate the relationships between the study variables (Figure 1). Data revealed a complex structure of interconnections between metabolic indicators, WHR and negative emotional symptoms. Strong connections (blue lines) were found between WHR with triglycerides and cholesterol, as well as moderate connections with glucose, while there is a moderate connection between cholesterol and triglycerides, while the former is weakly associated with glucose. Inspection of the network illustrates strong and complex interdependencies between emotional exhaustion, anxiety and depression symptoms and inverse with happy index. Furthermore, the network reveals positive connections between glucose and depression symptoms and weak connections with anxiety, which could indicate that glucose levels may positively influence depression and anxiety. A second possibility is that anxiety and depression increase the glucose.

The WHR has strong connections with metabolic indicators and moderate to weak connections with emotional states. This result is a fundamental bridging factor in this network. Secondarily, emotional exhaustion establishes bridging connections with WHR, but weak connections with cholesterol and glucose, that is, the higher the level of exhaustion, the higher the ICC but the lower the glucose levels. Furthermore, elevated levels of happiness are found in lower WHR, as well as a weak connection with low triglyceride levels and the presence of symptoms of burnout, depression, and anxiety.

Figure 2 shows the network’s strength, betweenness, and closeness. We then investigated the closeness of the nodes, cholesterol and triglycerides were identified as the two most important nodes of metabolic indicators and happiness as the most relevant in the emotional state of young people. In terms of strength, depression was statistically stronger from most of the other symptoms followed by emotional exhaustion. Regarding metabolic indicators, low cholesterol and glucose levels are favorable for the connections found in the network.

The betweenness nodes with the best index found were the WHI and happiness. The WHR acts as a bridge between metabolic indicators and emotional symptoms, as it is an important node in the connection that the other nodes have with each other. In the network, happiness is key because it is strongly and inversely connected to the factors of emotional discomfort and is the shortest path between the WHR and the other metabolic nodes. Finally, the structure of the network showed that the influence is exerted by the happiness node, followed by depression, so low levels of happiness will be affecting emotional stability and the fact that glucose, cholesterol and triglycerides increase.

## 7. Discussion

The present study aimed to determine the interaction of glucose, triglycerides, and cholesterol, along with BMI and WHR, in relation to emotional dysregulation in a sample of Mexican youths. The findings provide evidence of an interaction between biochemical indicators, anthropometric parameters and emotional distress in young individuals with varying body weight levels.

When analyzing the anthropometric results, it was found that most participants had a normal BMI, while only 14.9% were classified as individuals with obesity. This finding contrasts with previous studies [1,10] who reported higher prevalence rates of overweight and obesity in similar age groups. The difference may be attributed to the fact that the sample in the present study consisted of university students, who generally exhibit better anthropometric measurements and healthier eating habits

Regarding the WHR, 77.8% of participants were classified as low risk, in agreement with Noboa et al. [11], whose findings are consistent with those of our study. Research indicates that WHR is a key measure for assessing abdominal fat distribution. However, discrepancies were found when compared to the study by Çeltikçi et al. [5], which reported that 40% of their sample had a high WHR risk. This difference may be explained by sample characteristics such as the gender distribution, as those authors stratified their sample by male and female groups.

Regarding cholesterol, in our study, low levels were found, so differences were observed when compared to other studies [6] where elevated levels were more frequently found in populations with obesity. A possible explanation for this discrepancy is that, in the present study, a larger proportion of participants (48.6%) had a normal BMI and a waist-to-hip ratio indicative of low cardiovascular risk. Several studies have demonstrated that these anthropometric characteristics are associated with a more favorable lipid profile, including lower levels of triglycerides and LDL, and higher levels of HDL [56]. Within this context, maintaining healthy body weight and appropriate fat distribution may function as protective factors.

The glucose levels observed in the sample remained within normal ranges [42], which contrasts with previous studies [5] that reported a higher prevalence of hypoglycemia or glucose abnormalities in populations with similar characteristics, particularly among patients who have excess weight and obesity. This discrepancy may be explained by the fact that, in the present study, more than half of the participants had a BMI within the normal range, which may act as a protective factor [3].

The results obtained regarding the elevated levels of triglycerides are similar with previous studies [5,9,13]; which have documented that obesity and excess body fat are commonly associated with hypertriglyceridemia consistent with the findings in our study.

Analyzed as a whole, the results regarding cholesterol, triglycerides, and glucose reflect a favorable metabolic profile among the young individuals in our sample. Glucose levels remained within the normal range, which contrasts with previous studies [5,13], where a higher prevalence of dyslipidemia and glycemic alterations associated with excess weight was observed. However, partial agreement was found with studies that also report elevated triglyceride levels in young or overweight populations [5,13], which may suggest a metabolic susceptibility not exclusively dependent on BMI. The similarities between our results and previous studies [5,9,13] reinforce the pathophysiology of obesity.

Obesity, particularly visceral obesity, plays a role in the development of metabolic alterations through mechanisms such as insulin resistance, low-grade chronic inflammation, and endocrine dysfunction of adipose tissue, promoting the onset of dyslipidemia, such as increased cholesterol and triglyceride levels.

Regarding anxiety symptoms, the findings suggest that most participants (74.3%) exhibit mild to moderate manifestations of anxiety, while 6.8% experience severe symptoms. These results contrast with those reported in other studies conducted in similar populations [23,24,25]; which found a higher prevalence of elevated anxiety among patients who have excess weight. Regarding emotional exhaustion, the results indicate that most participants reported mild to moderate levels of stress, while only a minority presented with severe levels. This finding is consistent with previous studies [21] that recognize emotional exhaustion as a common condition during university years.

When analyzing the data on the levels of depression reported by the participants, 36.5% presented low symptomatology, and only 1.4% were classified as severe. These findings partially align with previous research [17] suggesting that depression is more prevalent among individuals with excess body weight. However, the low levels of severe symptomatology observed in our sample differ from earlier studies reporting a higher prevalence of depressive symptoms in overweight and obese populations [16,25].

The descriptive results of the present study on happiness show that average happiness levels were mean of 7.24, indicating a general tendency toward lower happiness, even among participants with normal weight. This finding partially aligns with previous reports by Saxena [27], who documented a negative correlation between obesity and happiness levels. However, in their study, lower happiness was predominantly observed among individuals with obesity, whereas in our sample, reduced happiness levels were also present among young individuals without excess weight.

The network analysis revealed a positive interaction between emotional exhaustion, depression, and anxiety, consistent with previous research [15,24]; which indicates that these three factors represent an interrelated emotional distress that tends to manifest jointly. Additionally, happiness showed an inverse relationship with symptoms of emotional exhaustion and depression [24,29]. The centrality nodes in the network structure corresponding to WHR, glucose, triglycerides, and cholesterol revealed a positive association, indicating that greater central adiposity is linked to alterations in the lipid profile [23,25].

The centrality analysis showed that happiness is the node with scores above 1; its interconnection reveals a negative association with metabolic indicators (glucose, triglycerides, cholesterol) and even with WHR, suggesting a potential protective role of emotional well-being in the metabolic health of university students. These findings are consistent with those reported by other authors [30,31]; who found that emotional well-being is related to better glycemic control and a lower risk of developing type 2 diabetes. Similarly, it has been documented that happiness is associated with lower levels of triglycerides [32], total cholesterol, and LDL cholesterol [33], reinforcing the idea that positive emotions contribute to a healthier lipid profile. Furthermore, higher levels of happiness have been linked to lower abdominal fat accumulation, consistent with the findings of Katsaiti et al. [28], who identified a relationship between happiness and stress in patients who have excess weight or obese. This association could be explained by the impact of happiness on the regulation of the hypothalamic–pituitary–adrenal (HPA) axis and cortisol levels, as well as by promoting healthy lifestyles that favor a more adequate distribution of body fat. However, our findings differ from those reported by Andrei et al. [26], who observed a decrease in happiness exclusively in individuals with grade III obesity (BMI > 40). This discrepancy could be explained by the sample characteristics, as our study primarily included young adults with normal weight (48.6%) and only 14.9% with obesity.

On the other hand, WHR showed positive relationships with indicators of emotional exhaustion, depression, and anxiety. Firstly, the relationship between WHR and emotional exhaustion can be explained by the physiological stress response, which increases cortisol levels, promoting abdominal fat accumulation and weight gain [20,21]. Similarly, emotional exhaustion acts as a risk factor for overweight and obesity [24].

The analysis of the WHR (waist-to-hip ratio) and depression suggests a bidirectional relationship, where emotional regulation may affect central adiposity, and conversely, central adiposity may increase depressive symptoms through inflammatory mechanisms and endocrine dysfunction. This pattern has been documented by Luppino et al. [17] and Jantaratnotai et al. [19], who report a consistent association between obesity and depressive symptoms. However, in the present study, more than half of the participants had a WHR within the normal range and a low percentage of obesity, which could have limited the presence of more severe depressive symptoms. The association between WHR and anxiety can be explained by the fact that anxiety promotes sustained physiological activation, that is, a prolonged state of alertness of the autonomic nervous system characterized by continuous elevation of stress-related hormones. This condition contributes to abdominal fat storage, as reported in studies conducted on overweight Mexican populations [23,25].

### 7.1. Practical Implications

The results of this study provide theoretical evidence of the interaction between metabolic indicators (glucose, cholesterol, and triglycerides), emotional distress (anxiety, depression, and emotional exhaustion), positive affect, and anthropometric measures in university students. The bidirectional interaction model proposed by Frank et al. [15] is partially confirmed, as significant associations between metabolic and emotional factors were observed even in the absence of severe obesity. Likewise, the data found in our study suggest that the effects of emotional distress on metabolism may manifest subtly even in the early stages of excess body weight, reinforcing the need to consider emotional factors as part of the metabolic risk profile in apparently healthy young populations.

According to the results obtained in this study, it is proposed that educational institutions implement a monthly follow-up plan for participating students. This implementation could be carried out by the faculties of Nutrition, Medicine, and Psychology. The Faculty of Nutrition would conduct monthly follow-ups, during which students would be measured, weighed, and have their anthropometric measurements taken in order to develop a personalized nutrition plan. The Faculty of Medicine would conduct quarterly follow-ups that would include blood pressure measurement, weight, height, and capillary blood sampling to assess glucose, cholesterol, and triglyceride levels. The Faculty of Psychology would conduct biweekly follow-ups, with one or two sessions during this period depending on the level of symptomatology the students present [49]. Brief techniques such as episodic future thinking [57] and emotional savoring [58] could be applied. If symptoms persist, the student would be referred for a more extensive psychological therapeutic process.

This strategy would allow for comprehensive and preventive support, aimed at reducing the risks of overweight, obesity, and emotional disturbances during the university period—an important and critical stage for the development of healthy habits. This type of early intervention could not only improve students’ quality of life, but also promote institutional-level prevention and comprehensive care, reinforcing the commitment to student well-being.

### 7.2. Limitations and Suggestions

The study has limitations that must be considered. First, the use of a cross-sectional design may limit the ability to establish relationships between metabolic indicators and emotional symptomatology. Future research with longitudinal designs could measure participants over time to observe how lipid alterations, anxiety, depression, emotional exhaustion, and happiness evolve. This would allow for the identification of whether certain network configurations predict the development of metabolic or mental health conditions. In addition, quasi-experimental studies could be conducted to verify whether the use of medication aimed at reducing metabolic indicators has an effect on improving emotional symptomatology.

A second limitation is the size (n = 78) and representativeness of the sample. Although the sample size is relatively small, it cannot be considered representative of the population of young Mexicans because participants were selected non-randomly and only from the central lowlands of Mexico, without including young people from various cities across the country. This reduces its representativeness, and thus, the results must be interpreted with caution.

The third limitation was difficulties in data collection, as participants did not attend scheduled appointments, which reduced the expected sample size.

The fourth limitation was identified in participants who did not meet the necessary conditions for biochemical measurements (e.g., arriving after an 8 h fast). Finally, a limitation refers to the fact that the biochemical analysis was performed via capillary puncture rather than peripheral venous extraction, which could cause variations in the results obtained.

### 7.3. Implications for Future Research

Our findings highlight the interaction between metabolic indicators and emotional symptoms, even in young individuals with normal weight. Therefore, as a future line of research, we consider two relevant approaches.

The first will involve conducting measurements using biomarkers (e.g., cortisol) and inflammation markers (e.g., cytokines). This will provide a deeper understanding of the underlying physiological mechanisms that connect emotional states with metabolic health.

The second approach focuses on the metabolic-emotional interaction. Based on our results, we will design specific intervention studies. We propose an intervention aimed at reducing emotional distress and depression or increasing happiness, to assess its impact on lipid alterations. Conversely, we propose designing dietary strategies that reduce adiposity and demonstrate improvements in participants’ emotional regulation. These studies could be designed to evaluate the implementation of such strategies on students’ health. Employing these different approaches will allow for a much clearer and more nuanced perspective on the relationship between metabolic health and emotional well-being, laying the groundwork for more effective and personalized interventions.

## 8. Conclusions

This research demonstrates the interaction between metabolic indicators and emotional symptoms in young individuals with both excess weight and normal weight. The results reveal that waist-to-hip ratio, triglycerides, and cholesterol occupy a central position within the metabolic cluster, while emotional exhaustion, anxiety, and depression form an emotional cluster that is directly related. Notably, happiness emerged as a central node with inverse connections to both metabolic and emotional risk factors, suggesting a protective effect on the participants’ overall well-being.

These findings highlight the importance of considering happiness not only as an indicator of well-being but also as a potential preventive modulating factor for metabolic risk [37,41]. Our results provide evidence supporting the need to include emotional symptoms in the early assessment of metabolic risk.

## Figures and Tables

**Figure 1 healthcare-13-03096-f001:**
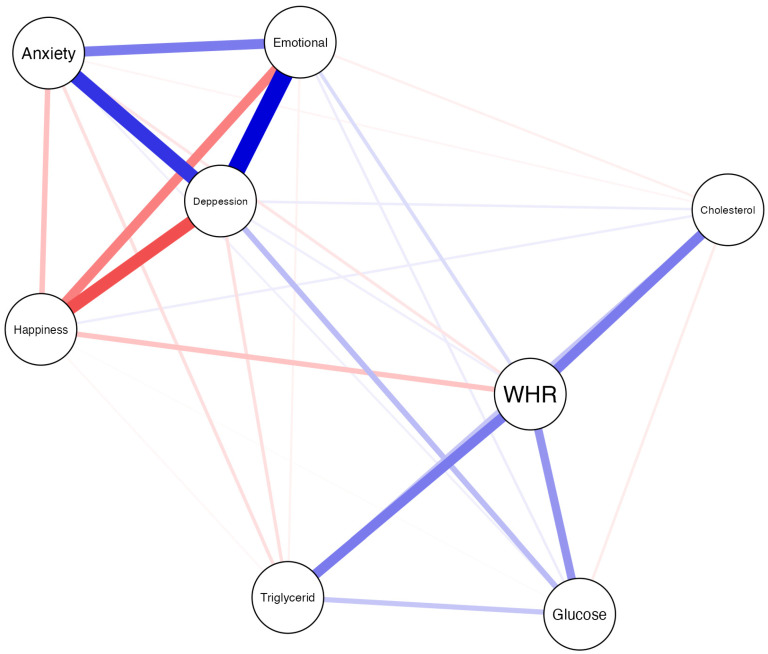
Network analysis of metabolic indicators, anthropometric measurement and emotional symptoms. The thickness of the lines indicates an increase in the level of relationship. Blue lines show positive correlations and red lines show negative correlations.

**Figure 2 healthcare-13-03096-f002:**
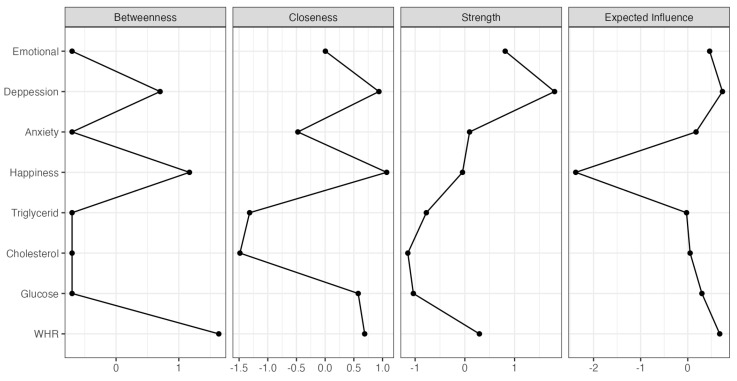
Centrality plot indices shown as standardized values z-score.

**Table 1 healthcare-13-03096-t001:** Descriptive statistics for metabolic and psychological factors.

	Mean	SD	Range	Category
Metabolic indicators				
Glucose	93.7	11.7	74–141	low *
Cholesterol	181	27.7	149–297	low *
Triglycerides	220	106	84–538	high limit
Anthropometric measures				
Weight	66.0	15.0	39.6–108	-
BMI	24.6	4.96	17.2–37.2	healthy *
Waist	78.3	11.1	58.9–82.0	normal
Hip	98	9.49	82–127	low *
Emotional factors				
Happiness	7.24	2.12	1–10	low
Depression	7.36	5.66	0–25	low
Anxiety	6.91	4.8	0–21	moderate
Emotional exhaustion	28.2	8.87	11–48	moderate

Note: * The categories were established using standard parameters, based on mean scores.

## Data Availability

The data presented in this study are available on request from the author. The data are not publicly available due to privacy and ethical restrictions.

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
