# Peer review of "Metabolic Indicators and Emotional Distress Symptoms Related to Overweight in Youth: A Brief Network Analysis"

_healthcare, 2025, doi:10.3390/healthcare13233096_

Round 1

Reviewer 1 Report

Comments and Suggestions for Authors

Thank you for an interesting and well written manuscript.  Minor changes can be made.  See attached document in Word.  The sample size was a bit low compared to what was determined as a representative sample.  In the methodology, to show validity, discuss the pilot study that was done to test the instruments.  In the discussion give more exact descriptions of the populations that this study is compared to as it may have a significant influence.

Author Response

Dear reviewer,

We appreciate your comments on the article and have made the changes you have indicated.

1)  The sample size was a bit low compared to what was determined as a representative sample.

Response: It is explained why the sample was smaller than initially proposed.

2) In the methodology, to show validity, discuss the pilot study that was done to test the instruments. 

Response: The instruments used are valid and reliable for the study sample. We add additional information on their psychometric evaluation.

3) In the discussion give more exact descriptions of the populations that this study is compared to as it may have a significant influence.

Response: In the discussion we incorporate the type of sample with which the information is compared

Reviewer 2 Report

Comments and Suggestions for Authors

The author's research explores the relationship between metabolic indicators and emotional symptoms in young individuals, both those with excess weight and those of normal weight. The findings show that the waist-to-hip ratio, triglycerides, and cholesterol are central components within the metabolic cluster. Concurrently, emotional exhaustion, anxiety, and depression form an emotional cluster that is directly related to these metabolic indicators.

Interestingly, happiness emerged as a key factor, demonstrating inverse connections to both metabolic and emotional risk factors. This suggests that happiness may play a protective role in the overall well-being of the participants.

The authors emphasize the importance of viewing happiness not only as an indicator of well-being but also as a potential preventive factor for metabolic risks. Their results underscore the need to include an assessment of emotional symptoms in the early evaluation of metabolic risk.

The strength of the manuscript: a modern way of connecting (network analysis) psychological and biochemical parameters with obesity in young people.

Weakness of the manuscript: some physiological parameters were not monitored, lack of information on the use of drugs.

Comments and suggested correction:

Blood pressure is usually increased with stress and anxiety. The authors should have measured blood pressure in the examined individuals and included hypertension in the network analysis. The authors should explain in a few sentences in the manuscript why the blood pressure value was omitted from the analysis.

It may be important if the person being examined uses any medication against depression, stress, etc.

It may also be important to know whether the person examined is applying any therapy for obesity.

Author Response

Dear revisor,

We appreciate your comments on the document for improvement. We have made the improvements you suggested.

Comments: lack of information on the use of drugs.

Response: One of the exclusion criteria was the use of any type of medication or drug consumption.

Comments: Blood pressure is usually increased with stress and anxiety. The authors should have measured blood pressure in the examined individuals and included hypertension in the network analysis. The authors should explain in a few sentences in the manuscript why the blood pressure value was omitted from the analysis.

Response: Blood pressure was not shown to be associated with emotional exhaustion (r = - .02 ; p = 0.43) or anxiety (r = .020 ; p = 0.86)  for this reason it was excluded from the network analysis.

Comments: It may be important if the person being examined uses any medication against depression, stress, etc.

Response: One of the exclusion criteria was the use of any type of medication or drug consumption.

Comments: It may also be important to know whether the person examined is applying any therapy for obesity.

Response: For future studies we will ask this question, since at present there is no data on the subject.

Reviewer 3 Report

Comments and Suggestions for Authors

The article addresses an interesting topic. Although several studies on nutritional neuroscience are already known, any further contributions are always useful. However, certain considerations are made for the improvement of the article:

1.- It is advisable to increase the number of keywords for better identification of the article.

2.- What happened to reduce the initial sample to the final sample included in the study? Is there any intrinsic variable not included in the exclusion criteria?

3.- Why does the sample consist only of university students studying health sciences? Are there any criteria or variables of interest?

4.- It would be necessary to detail the instruments used for metabolic measurements.

5.- Why were three non-specific metabolic parameters used, given that they are highly susceptible to hormones that can affect them and no hormone studies have been conducted?

6.- Clarification regarding the results, as it is not clear whether there is a bidirectional relationship between metabolic parameters, anxiety and depression.

7.- The bibliography contains some older citations (6, 13, 15, 19, 24, 30, 54) that could be updated with regard to the subject matter studied to allow for a better discussion.

Author Response

Dear reviewer, we appreciate your comments on how we can improve the article. We respond to your comments below.

Comments: 1.- It is advisable to increase the number of keywords for better identification of the article.

Response: we incorporated additional keywords.

2.- Comments: What happened to reduce the initial sample to the final sample included in the study? Is there any intrinsic variable not included in the exclusion criteria?

Response: The difference between the calculated sample size (n=217) and the final number of participants included in the study (n=78) was due to various operational and contextual factors that affected the data collection process. The main reasons for this were:

  1. Despite meeting the inclusion criteria, a significant proportion of students voluntarily declined to participate, limiting effective recruitment.
    2. The study was conducted during different academic periods, some of which were heavily attended, making it difficult to coordinate sampling sessions and administer the instruments.
    3, Several cases were discarded for not meeting specific requirements, such as signing an informed consent form, the presence of medical conditions that could bias the results, and, most importantly, the lack of complete data.

3. Comments - Why does the sample consist only of university students studying health sciences? Are there any criteria or variables of interest?

Response: The selection of university students in health sciences as a sample primarily responds to specific methodological and logistical criteria that guarantee the relevance and validity of the study. First, this population group has homogeneous characteristics in terms of educational level, training context, and exposure to health-related content, which allows for controlling confounding variables and improving the accuracy of the analyses.

The selection of university students in health sciences as a sample primarily responds to specific methodological and logistical criteria that guarantee the relevance and validity of the study. First, this population group has homogeneous characteristics in terms of educational level, training context, and exposure to health-related content, which allows for controlling confounding variables and improving the accuracy of the analyses.

Furthermore, health science students represent a population of strategic interest, given that they are training to become professionals who will directly influence health promotion, prevention, and care in their communities. The inclusion criteria were: enrollment in a health science program, at least one semester completed, and voluntary participation in the study.

5.- Comments: Why were three non-specific metabolic parameters used, given that they are highly susceptible to hormones that can affect them and no hormone studies have been conducted?

Response: The selection of glucose, cholesterol, and triglycerides as metabolic parameters in the present study reflects their widely recognized clinical value as primary indicators of overall metabolic status. These biomarkers are essential components of the basic metabolic profile and allow for the identification of early alterations associated with chronic diseases such as type 2 diabetes, dyslipidemia, and cardiovascular disease.

While it is true that these parameters can be influenced by hormonal factors—such as thyroid activity, insulin secretion, or the hypothalamic-pituitary axis—their diagnostic usefulness is not negated by the absence of complementary endocrinological studies. On the contrary, their sensitivity to hormonal changes can be interpreted as an advantage, as it allows for the indirect detection of systemic imbalances that could justify further, more specific studies.

The methodological decision not to include hormonal studies is based on the exploratory approach of the study, which focuses on establishing metabolic risk patterns in a specific population. Furthermore, the accessibility, low cost, and standardization of glucose, cholesterol, and triglyceride tests make them practical and reproducible options for population studies, especially in contexts where resources for hormonal analysis are limited.

Future research is considering including hormonal markers such as insulin, cortisol, and melatonin to broaden our understanding of endocrine-metabolic interactions and strengthen the predictive capacity of the proposed model.

6. Comments:  Clarification regarding the results, as it is not clear whether there is a bidirectional relationship between metabolic parameters, anxiety and depression.

Response: In brief, our findings suggest a bidirectional relationship between metabolic parameters and emotional symptoms, as central adiposity, as determined by WHR, cholesterol, and triglycerides, was found to be associated with symptoms of depression, anxiety, and emotional exhaustion. Emotional distress can negatively influence the metabolic profile through mechanisms such as the onset of chronic inflammation. Happiness, for its part, acts as a protective factor, suggesting that a positive emotional state could reduce metabolic risks. In the discussion we include a brief paragraph summarizing our findings.

7 Comments.- The bibliography contains some older citations (6, 13, 15, 19, 24, 30, 54) that could be updated with regard to the subject matter studied to allow for a better discussion.

Response: We incorporate old references to show that there are previous precedents that address the topic and that we would approach it from a new perspective.

If necessary, we will reduce old references in a second round of review.

Round 2

Reviewer 3 Report

Comments and Suggestions for Authors

 I agree with your responses. I think you must reduce old references in a second round of review.

Author Response

Dear reviewer,

we have removed from the article any old references or those that do not contribute in-depth to our study.

We are waiting for any additional comments.

Best regards.